# Antimicrobial efficacy and biocompatibility of extracts from *Cryptocarya* species

**Jacqueline de Oliveira Zoccolotti**[1☯], **Alberto José Cavalheiro**[2☯], **Camilla Olga Tasso**[1☯], **Beatriz Ribeiro Ribas**[1☯], **Túlio Morandin Ferrisse**[1☯], **Janaina Habib Jorge**[1☯]*

**1** Department of Dental Materials and Prosthodontics, School of Dentistry, São Paulo State University (Unesp), Araraquara, São Paulo, Brazil, **2** Department of Biochemistry and Organic Chemistry, Institute of Chemistry, São Paulo State University (Unesp), Araraquara, São Paulo, Brazil

☯ These authors contributed equally to this work.
* habib.jorge@unesp.br

**Data Availability Statement:** Data are available in: http://hdl.handle.net/11449/214657.

**Funding:** The author Jacqueline de Oliveira Zoccolotti received a doctoral scholarship by the

## Abstract

This study evaluated the efficacy of *Cryptocarya* spp extracts on biofilm of *Candida albicans* and its biocompatibility. Mature biofilm of *C. albicans* was formed on denture base acrylic resin samples and the fungicidal effect of the extracts was evaluated by Alamar Blue® assay, counting colony-forming units (CFU/mL) and confocal laser scanning microscopy (CLSM). Cytotoxicity of extracts from *Cryptocarya* species was evaluated by AlamarBlue® assay, using normal oral keratinocytes (NOK) cells. In additional, Analysis of plant extracts by ultra-high-performance liquid chromatography–diode array detector–tandem mass spectrometry (UPLC-DAD-MS) was performed. The results showed significant reduction in the cellular metabolism and in the number of CFU/mL of *C. albicans* (p<0.05). The concentration of 0.045 g/mL completely inhibited the number of CFU/mL. Regarding cytotoxicity, all extracts decreased cell viability compared to the control group. CLSM analysis showed predominance of live cells, but with a great difference between the groups. Antimicrobial activity of extracts from *Cryptocarya* on *C. albicans* biofilm was confirmed. However, all extracts showed toxicity on NOK cells.

## Introduction

Oropharyngeal candidiasis is an opportunistic infection of the mucosa caused by *Candida* species [1, 2] and *Candida albicans* is the most common etiological agent [3, 4]. One such example of oropharyngeal infection is Denture Stomatitis (DS). DS is an inflammatory condition encountered most commonly in removable denture wearers and is characterized by diffuse erythema in the areas covered by the dentures, which can present as reddened spots, different textures in the palatal mucosa surface or with a homogeneous aspect [5–7]. The etiology of DS is extremely variable and is considered multifactorial. Thus, *Candida* spp. infection, trauma, gender, nutritional deficiency, poor fitting dentures, continual wearing, reduced salivary flow, lack of hygiene are some of the factors involved in the DS etiology [8–12]. However, the presence of *Candida* ssp. biofilm remains one of the most important factors for the development of this condition.

Coordenação de Aperfeiçoamento de Pessoal de Nível Superior – Brasil (CAPES) – Finance Code 88887.513643/2020-00. The funders had no role in study design, data collection and analysis, decision to publish, or preparation of the manuscript.

The capacity of *C. albicans* to form structured biofilm helps induce infection and makes the treatment more difficult because of the better resistance of its cells against chemical and physical aggressions [13]. Additionally, the treatment of DS is complex due to its multifactorial etiology. The most common treatment is the use of topic and systemic antifungal agents associated with methods of hygiene and disinfection of dentures such as chemical disinfection, microwave irradiation and, even the change of the denture [14]. Currently, the use of antifungal therapies must be limited because of their toxicity, low rates of effectiveness, induction of fungal resistance and low solubility [15]. Therefore, many current researches seek alternative therapies for the treatment of infections, with phytotherapy being a viable alternative to achieve this goal.

In the southeastern region of Brazil, more specifically in the Atlantic Forest region, seven species of *Cryptocarya* are found. Among the existing species, studies with the ethanolic extract of the pulp of green fruits and seeds of *C. moschata* demonstrated the existence of styrylpirones, which showed antifungal activity [16]. Besides that, in other studies, leaves obtained from *C. moschata* and *C. mandioccana* after being isolated, showed also antifungal activity [17]. Despite the results found, to the extent of the authors knowledge, there are few studies in the literature evaluating the activity of *Cryptocarya* species on *C. albicans* biofilm, aiming to promote a low-cost, accessible and non-synthetic option for the treatment of oral fungal infections such as DS. That being said, the aim of this study was to evaluate the efficacy of extracts from *Cryptocarya* species on biofilm of *Candida albicans* formed on samples of denture base acrylic resin and its cytotoxicity over normal oral keratinocytes. The null hypothesis of the study was that the *Cryptocarya* species would not be effective against *C. albicans* biofilm and would not be cytotoxic to NOK cells.

## Materials and methods

### Plant material

*Cryptocarya moschata* Nees & Mart. (Lauraceae) leaves were collected from the grove of Camará Mudas Florestais (21°56'22.2"S, 47°59'30.6"W), Ibaté-SP, Brazil, in June 2015, and fruits were collected from the Bosque dos Alemães (22°53124.2"S, 47°04'07.0"W), Campinas-SP, Brazil, in December 2018. *Cryptocarya mandioccana* Meisn. and *C. saligna* Mez (Lauraceae) leaves and fruits were collected from Serra do Mar State Park, Picinguaba Nucleus (23°20'20.8"S, 44°50'08.3"W), Ubatuba-SP, Brazil, in February 2019. Plant material were identified by Prof. Dr. Pedro L. R. de Moraes and a voucher specimen of *C. moschata* (Moraes 2347) and of *C. saligna* (Moraes 2472) were deposited in Escola Superior de Agricultura Luís de Queiroz herbarium (ESA, Piracicaba-SP, Brazil) and a voucher specimen of *C. mandioccana* (NP0001) was deposited in Herbarium Maria Eneyda P. Kaufmann Fidalgo (São Paulo, SP, Brazil). Authorization for harvesting the plant material was granted by Instituto Florestal, Secretaria do Meio Ambiente do Estado de São Paulo (SMA010.111/2018) and permission to access the Brazilian genetic patrimony was provided by SISGEN (A375D64).

### Preparation of plant extract

First, the collected samples were dried in an oven with ventilation and air renewal, for 48 hours, at a temperature of 40°C. Then, they were pulverized in a knife mill. The plant extracts were prepared by sonication (ultrasound-assisted extraction-UAE) of the plant material with hydroalcoholic solvent (water 30% and ethanol 70%). Three extractions were carried out with the solvent by UAE, for 20 minutes each, in order to obtain extracts of leaves, trunk, bark and fruit of each *Cryptocarya* species, using 1:10 ratio (wt v-1) of raw material and solvent. Then,

the extract was filtered and dried until complete solvent elimination. For this, the ethanol was removed in a rotavapor and elimination of water by lyophilization [18].

## Analysis of plant extracts by UPLC-DAD-MS

Around 10 mg of hydroalcoholic extracts were dissolved in 2 mL of ethanol/water 1:1 (v/v), centrifuged at 5000 g by 5 min. The supernatant was transferred to 1.5 mL vials and subjected to ultra-high-performance liquid chromatography–diode array detector–tandem mass spectrometry (UPLC-DAD) analysis on a DIONEX 3000 ultimate RS system, on line with an Amazon IT Mass Spectrometer (MS). The chromatographic analysis was performed on a Kinitex® XB-C18 column (50 x 2.1 mm, 2.6 μm, 100 Å, Phenomenex) using as mobile phase ultrapure water (A) (Milli Q system) and ethanol (B) 5(HPLC grade), both with 0.1% of formic acid, in gradient mode (5 to 60%B in 20 min) at a flow rate of 0.5 mL/min. Additional step from 60 to 100% B in 10 min and holding in 100% B for 5 min was used to clean up the column after each sample analysis. UV/Vis spectra were acquired from 200 to 600 nm and chromatograms recorded in 280 nm. The MS data were acquired in positive and negative mode, using an amaZon-SL electrospray ionization ion trap (ESI-IT) Bruker Daltonics spectrometer. The data were processed through Bruker Compass Data Analysis 4.1 software (Bremen, Germany).

## Minimum Inhibitory Concentration (MIC) and Minimum Fungicidal Concentrations (MFC)

The microorganism used in this study was an American Type Culture Collection strain of *C. albicans* (ATCC 90028). The cells were stored in a freezer at -80° C until the tests were carried out. For the tests, 25 μL of *C. albicans* were seeded in Petri dishes containing the culture medium Sabouraud Dextrose Agar (SDA) with 5% chloramphenicol, and the dishes were incubated at 37°C for 48 hours. Yeast colonies were inoculated in 10 mL of Yeast Nitrogen Base (YNB) and grown overnight aerobically at 37°C for 16 hours. Then, 0.5 mL of the initial culture was transferred to 9.5 mL of fresh YNB. The optical density was evaluated and the culture was cultivated aerobically at 37°C for 8 hours. To obtain standard suspensions of *C. albicans*, each culture was harvested after centrifugation at 4000 rpm for 5 minutes, washed with phosphate buffered saline (PBS) twice and resuspended in the Roswell Park Memorial Institute medium (RPMI) at $10^6$ units colony forming per milliliter (CFU/mL), adjusting the optical density of the suspension to 540 nm. The MIC values were determined by the static incubation of the suspensions of the microorganism, for 24 hours at 37°C, exposed to the different concentrations of the *Cryptocarya* extracts (100%, 50%, 25%, 12.5%, 6, 25%, 3.12%, 1.56%, 0.78%). For this, the crude extracts were weighed and aliquots and, then, diluted in phosphate buffered saline (PBS) with ethanol (5%) in the proportions of interest. After adjusting the concentrations, 10 μL aliquots of the suspension obtained for *C. albicans* were placed in a 96-well plate, and 200 μL of the dilutions of each extract were added to each well, in triplicate. The plate was incubated at 37°C for 24 hours. The microbial growth in each well was analyzed visually to determine the MIC. Wells without visual growth were seeded on SDA to determine MFC and were placed in an incubator at 37°C for 24 hours. After that period, CFUs were counted. MFC was confirmed in the well where there was no growth of microorganism. The analyzes were carried out in triplicate on 3 different occasions.

## Determination of the Minimum Fungicidal Concentration of cells in biofilm (MFCb)

For biofilm formation, first 100 μL of the culture medium, containing the fungal cells at a concentration of $1 \times 10^6$ cells/mL, supplemented with 100 μL of buffered RPMI-1640 medium, with

3-(N-Morpholino) propane sulfonic acid (MOPS) and pH 7.0, were placed at each well of a 96-hole plate. Then, the plate was taken for incubation at 37˚C, in an orbital shaker (75 rpm) for 90 minutes (initial adhesion). Subsequently, the non-adherent cells were removed by washing with sterile PBS buffer twice, and 200 μL of RPMI-1640 culture medium buffered with MOPS (pH 7.0) were placed in each well of the plate to biofilm formation. The plate was taken for incubation for 24 hours at 37˚ C under orbital emission (75 rpm). After 24 hours, 100 μL of medium was removed from the wells and the same volume of RPMI-1640 buffered with fresh MOPS (pH 7.0) was added. The plate was then incubated again for another 24 hours, totaling 48 hours of mature biofilm formation. After 48 hours, the medium was aspirated and the non-adherent cells were removed by washing the biofilms twice with sterile PBS buffer. After the formation of the biofilm, 200 μL of the solutions of *Cryptocarya* extracts were added to each well in the plate in different services, starting from the MIC concentration for planktonic cells and increasing each well gradually by 5x, 10x, 15x, until the concentration maximum possible (100%). These solutions were kept in contact with the biofilm for a period of 24 hours at 37˚C. After the incubation period, a MFCb was provided as the lowest concentration to inhibit the growth of the biofilm, which was observed by the method of counting colony forming units. The analyzes were carried out in triplicate on 3 different occasions.

## Fabrication of acrylic resin specimens

Disc-shaped samples (14 mm x 1.2 mm) of a denture base acrylic resin (Vipi Wave, Dental Vipi Ltda) were fabricated and polymerized according to the manufacturer's recommendations. After polymerization, excess resin from the processing was removed using a sterilized trimming bur and the specimens were stored in distilled water for 48 hours (zero time) at 37˚C to remove excess monomer. After storage for 48 hours, the surface roughness was measured to standardize the specimens and to homogenize the groups. The specimens that had roughness between 2.7 and 3.7 μm were selected for the following tests [19].

## Biofilm formation on samples

*C. albicans* cultures were performed as described for MICs. For the analysis of the capacity of biofilm formation, specimens from each experimental group (n = 9) were placed individually in a 24-well plate, and on each sample, 2 mL of the filtered saliva (CAAE: 79249417.0.0000.5416) was added to each orifice and the plate was incubated in an oven at 37˚C for 90 minutes to form the salivary pellicle. Then 750 μL of the aliquot of RPMI medium with *C. albicans* was added to each sample and incubated for 90 minutes at 37˚C (adhesion phase) under agitation at 76 rpm. After that time, liquid cultures were removed from the wells. Subsequently, the non-adherent cells were removed from the samples by washing with sterile PBS buffer twice, and 1500 μL of RPMI-1640 culture medium buffered with MOPS (pH 7.0) was placed in each well of the plate for biofilm formation. The plate was taken for incubation for 24 hours at 37˚C under orbital shaking (75 rpm). After 24 hours, 750 μL of medium were removed from the wells and the same volume of RPMI-1640 buffered with fresh MOPS (pH 7.0) was added. The 24-well plate, containing the samples, was then incubated again for another 24 hours, totaling 48 hours of formation of a mature biofilm. After 48 hours, the medium was aspirated and the non-adherent cells were removed by washing the biofilms twice with sterile PBS buffer.

## Antifungal activity of extracts of *Cryptocarya* species and experimental groups

After the formation of the mature biofilm (48 hours) on the acrylic resin samples, each well of the 24-hole plate, containing the samples, was filled with 1 mL of the following solutions:

extracts of *Cryptocarya* species solution (*C. moschata* and *C. mandioccana*) in the MFCb (previously determined); nystatin solution at 100,000 IU / mL (positive control) and phosphate-buffered saline (PBS) solution (negative control). The solutions remained in the wells for a period of 60 minutes. Then, the samples were washed twice with PBS to remove the excesses of the different solutions and the non-adhered cells. To assess the antimicrobial effect of the different solutions, tests for the evaluation of cellular metabolism (Alamar Blue® assay) and counting the number of viable colonies (CFU / mL) were performed. The analyzes were carried out in triplicate on 3 different occasions.

## Alamar Blue® assay

The cellular metabolism of the *C. albicans* biofilm was evaluated using the Alamar Blue® assay, a technique that measures cell viability through the activity of mitochondrial enzymes. After the contact period of the samples with the solutions of each experimental group, they were washed twice with PBS and 1500 μL of new culture medium was placed in each hole in the plate. Then, 150 μL of Alamar Blue® solution was added. The plates were then placed in the orbital shaking incubator at 37˚C and 76 rpm. After 4 hours, the fluorescence reading was performed. The fluorescence of the samples was measured using Fluoroskan Ascent at 560 nm (A560) and 590 nm (A590).

## Counting the number of viable colonies (CFU / mL)

For the quantification of microorganisms, the samples containing the *C. albicans* biofilm were transferred to Falcon tubes containing 4.5 mL of pure PBS and vortexed for 1 minute to detach the cells. Then, the serial dilution process was performed in Eppendorf tubes containing 900 μL of pure PBS by transferring 100 μL aliquots of the original solution. Dilutions from $10^{-1}$ to $10^{-4}$ were obtained. Two 10 μL aliquots of the $10^{-3}$ and $10^{-4}$ dilution were dripped onto Petri dishes containing SDA. Then, all plates were incubated at 37˚C for 48 hours. After this period, the plates were placed on a manual colony counter and the number of colonies was determined. The number of colony forming units per milliliter (CFU/ mL) was calculated according to the following formula: CFU / mL = number of colonies x $10^n$/ q. In this formula, n is equivalent to the absolute value of the chosen dilution (from 3 to 4) and q is equivalent to the quantity, in mL, seeded from each dilution in the plates (0.010 mL = 10 μL).

## Confocal laser scanning microscopy (CLSM)

The specimens of each experimental group were placed in a Petri dish (5 cm in diameter) and 10 mL of saline solution containing 2 mL of Live/Dead dye (Molecular Probes, Eugene, OR, USA), containing SYTO-9 and propidium iodide (PI) (Molecular Probes, Inc., Eugene, OR, USA), were added carefully over the samples. The plates were incubated for 15 minutes in the dark. The maximum excitation and emission used were approximately 480/500 nm for SYTO-9 and 490/635 nm for PI. Images were obtained with a confocal microscope (Leica Microsystems GmbH, Wetzlar, Germany).

## Cytotoxicity assay

The cytotoxicity effect of the *Cryptocarya* extracts was assessed by the cell culture method. Normal oral keratinocytes (NOK) cells provided by Prof. Dr. Carlos Rossa Junior (Department of Periodontics, Faculty of Dentistry of Araraquara—UNESP, Brazil) were used. The cells were propagated in the Dulbecco's Modified Eagle's Medium (DMEM) with 7.5% fetal bovine serum and 80 μg/mL gentamicin. The culture was maintained at 37˚C in an atmosphere of 5% $CO_2$ and 95% air. Next, 10 μL of the cells ($1 \times 10^5$ cell/mL) were added to 90 μL Trypan Blue. Of this

solution, 10 μL were removed and introduced into a Neubauer hemocytometer counting chamber, in which the viable cells were counted with an optic microscope (40x magnification).

After that, the cells ($1 \times 10^4$ cell/mL) were seeded into 96-well culture plates and incubated for 24 hours at 37˚C in the same atmosphere. After this period, the culture medium was removed and 100 μL of culture medium with the medicinal plant solution in the previous selected concentrations were added in each well. Subsequently, 10 μL of the Alamar Blue® solution were also added. After 6 hours, the first fluorescence reading was performed, the second reading after 12 hours and the third reading after 24 hours. The fluorescence of the samples was measured using Fluoroskan Ascent at 560 nm (A560) and 590 nm (A590). The cytotoxicity assay was performed in triplicate on 3 different occasions.

## Statistical analysis

In order to evaluate if the extracts interfered in the reduction of the biofilm, the CFU/mL values were transformed into logarithm ($\log_{10}$) and the Kruskal-Wallis and Dunn's post hoc tests were carried out. To evaluate the fluorescence values data, the analysis of variance (ANOVA) followed by the Tukey post-test was used. The level of significance used was 5%.

In the quantitative analysis of CLSM, it was obtained the ratio (R) between alive/dead, as a function of the fluorescence intensity emitted by the sample. When the ratio R is greater than 1, it means that there is a predominance of living cells, if R equals 1, there are 50% alive and 50% dead, and if R is less than 1, there is a predominance of dead cells.

In addition, for the cytotoxicity assay a qualitative and quantitative analysis were made separately. Regarding the quantitative analysis, the number of viable cells were submitted to normality and homogeneity tests. Then, the TWO-WAY mixed ANOVA followed by median estimate with 95% confidence interval and Tukey post-test were carried out ($\alpha = 0.05$). Moreover, the *Cryptocarya*. extracts solutions were classified according to the cytotoxic effect (qualitative analysis) as the following description: 0: no cytotoxic (inhibition less than 25% in relation to control group); 1: Quietly cytotoxic (inhibition between 25% and 50% in relation to control group); 2: Moderately cytotoxic (inhibition between 50% and 75% in relation to control group) and 3: Intensely cytotoxic (inhibition more than 75% in relation to control group).

## Results

### Chemical analysis

The UHPLC-DAD-MS analysis (Fig 1) revealed a common flavonoid profile to the three plant species, based on quercetin-3-O-glicosides (3–7) and trimethylquercetin (11) (Fig 2). However, the aporphine alkaloids menisperine (1) and xanthoplanine (2) were detected in leaves and fruits of *C. moschata* and *C. mandioccana*, but not in leaves of *C. saligna*. Additionally, styrylpyrones have a more restrict occurrence, once the goniothalamine (8) and 6-styryl-2-pyrone (9) were founded just in *C. moschata*, while deacetylcryptocaryalactone (10) was detected only in *C. mandioccana*. The compounds annotation was based on the UV and MS spectra and in previous studies [18, 20, 21].

### Determination of Minimum Inhibitory Concentration (MIC) and Minimum Fungicidal Concentration (MFC) for planktonic cells

The test for the analysis of the MIC for planktonic cells was performed for each of the extracts of leaves, fruits, peels and trunks of *C. moschata*, *C. saligna* and *C. mandioccana*, on three different occasions. For the species *C. moschata* and *C. mandioccana* in the extracts obtained from the leaves and fruits, the same MIC value was found as a result, this being at the dilution

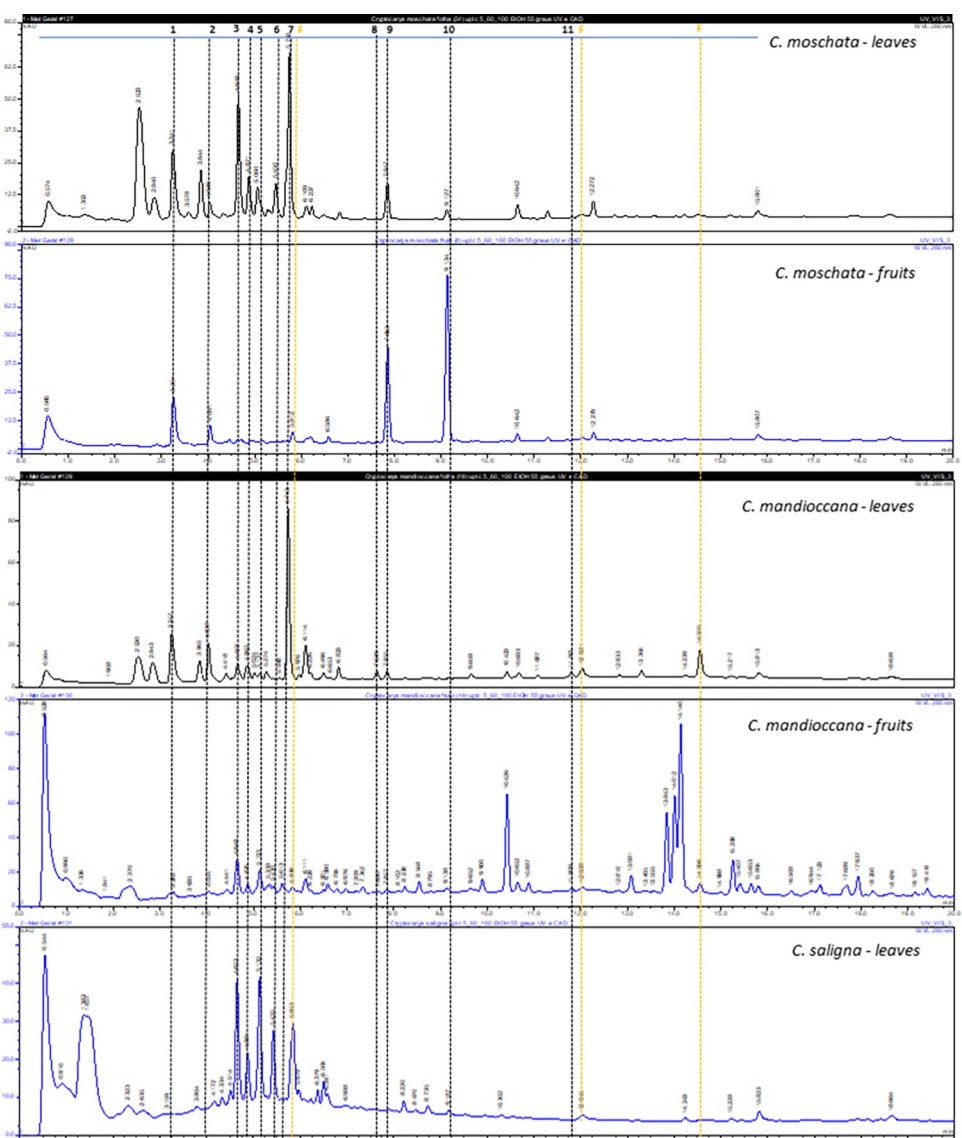

**Fig 1. UHPLC-DA chromatograms from the *Cryptocarya* extracts, plotted at 280 nm.** Chromatographic conditions. Kinitex® XB-C18 column (50 x 2.1 mm, 2.6 μm, 100 Å, Phenomenex), eluted with ultrapure water (A) and ethanol (B), both with 0.1% of formic acid, in gradient mode (5 to 60%B in 20 min) at a flow rate of 0.5 mL/min. Numbers 1–11 refer to identified compounds. F = unidentified flavonoids.

of 0.003 g / mL (MIC was the concentration that eliminated 100% of planktonic cells of *C. albicans*). For the extracts of leaves and fruits of *C. saligna*, even increasing the concentration to 0.015 g / mL, no efficacy was found against the cells of *C. albicans*, as well as the extracts obtained from the bark or trunk of the three species, which did not show antifungal activity at this concentration and therefore were eliminated from the next tests of this study.

## Determination of the Minimum Fungicidal Concentration of cells in biofilm (MFCb)

The MFCb was equal to 15 times the MIC value, i.e 0.045 g / mL. This concentration for the extracts of leaves and fruits of the species of *C. mandioccana* and *C. moschatta*, after 1 hour of contact, completely inhibited the biofilm formed on the bottom of wells of a 24-hole plate.

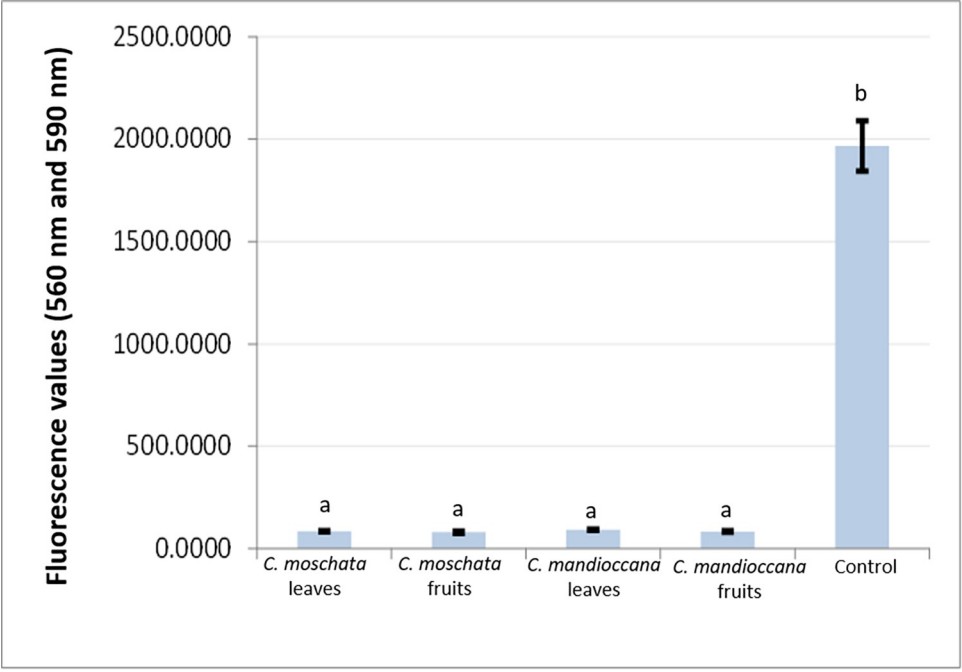

**Fig 2. Molecular structures of the main compounds identified in the *Cryptocarya* extracts.**

## Assessment of cellular metabolism using the Alamar Blue® test

The cellular metabolism of *C. albicans* was evaluated, first, in MFCb, equivalent to 15 times the MIC value (Fig 3). In addition, to complement the results, the 10-fold concentration of MIC was also assessed (Fig 4). The results showed that, for both concentrations, there was a statistical difference between the tested extracts and the control group, with a significant reduction in cellular metabolism (p<0.05).

**Fig 3. Fluorescence values of *C. albicans* cells submitted to treatment with *Cryptocarya* extracts at a concentration of 0.030g/mL (10 times the MIC value).**

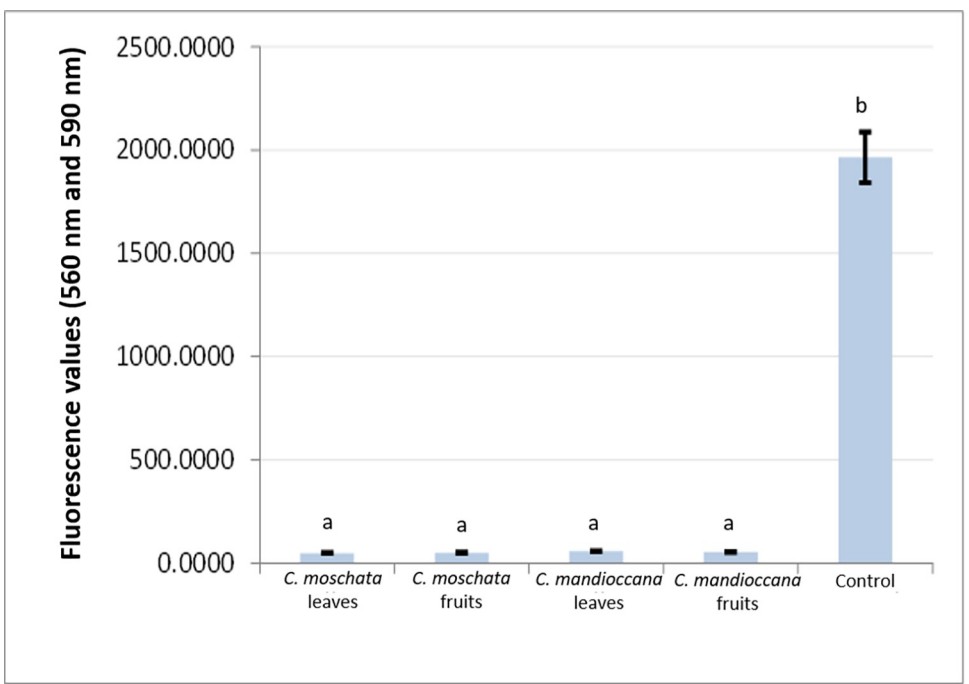

**Fig 4. Fluorescence values of *C. albicans* cells submitted to treatment with *Cryptocarya* extracts at a concentration of 0.045 g/mL, 15 times the MIC value.**

## Counting the number of viable colonies (CFU / mL)

The number of CFU / mL of *C. albicans* was evaluated at concentrations equivalent to 10 times the MIC value (0.030 g/mL) and at the 15-fold MIC concentration (0.045 g / mL). The concentration of 0.045 g/mL completely inhibited cell growth (MFCb), so no statistical analysis was performed on these data equal to 0 (Fig 5). Regarding the concentration of 10x to MIC, the results showed that, for the experimental groups, there was a reduction of 1 log in the number of CFU/mL in relation to the negative control group, but without a statistically significant difference. Statistical difference was found only between the negative control group and the positive control group (100,000 IU/mL nystatin solution) (Fig 6).

## Confocal laser scanning microscopy (CLSM)

The CLSM images were showed in the Fig 7. The Fig show *C. albicans* adhered to acrylic resin samples after exposure to the extracts (10 and 20 min) and nystatin (10 and 20 min) compared with the negative control group. Image capture was set for simultaneous observation of both green and red fluorescence. The quantitative analysis showed predominance of live cells, but with a great difference between the control group (R = 10.8) and the other groups: extract 10 minutes (R = 4.6), extract 20 min (R = 2.2), nystatin 10 min (R = 2.1) and nystatin 20 min (R = 3,4).

## Cytotoxicity assay

The results showed that there was statistically significant difference between the groups, at all times evaluated (Fig 8). It was observed that all extracts decreased cell viability compared to the control group. Furthermore, it was observed that there was an increase in the number of cells with time. When the results were converted into percentages and compared with the

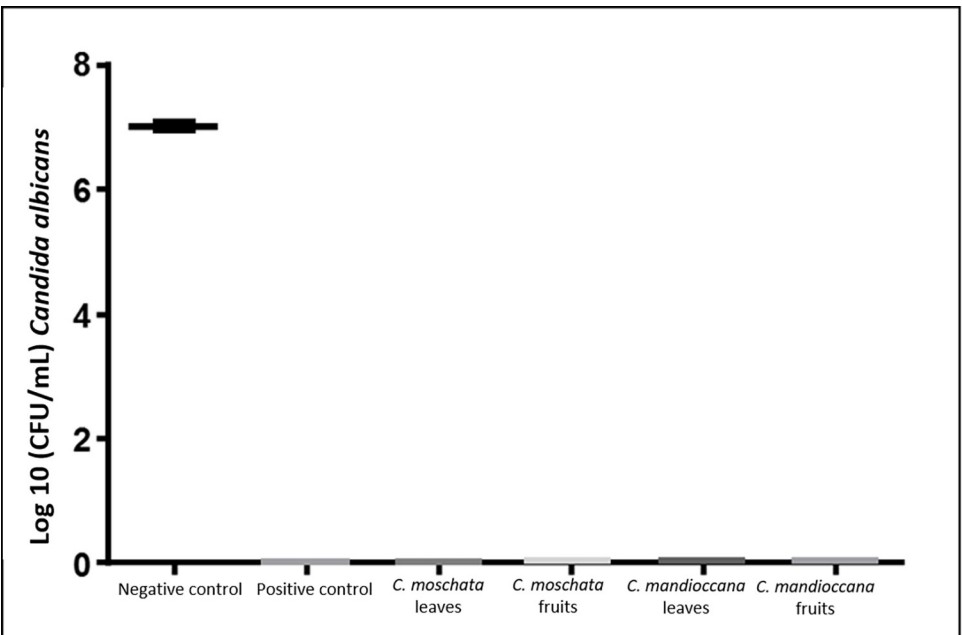

**Fig 5. CFU/mL values of *C. albicans* cells submitted to treatment with *Cryptocarya* extracts at a concentration of 0. 045 g/mL (15 times the MIC value).**

control group (Fig 9), it was found that the extracts in the MFCb (0.045 g / mL) were considered intensely cytotoxic, at all times. In MICs (0.003 g / mL), extracts ranged from non-cytotoxic to intensely cytotoxic. For all groups, the percentage of viable cells decreased over time compared to the control.

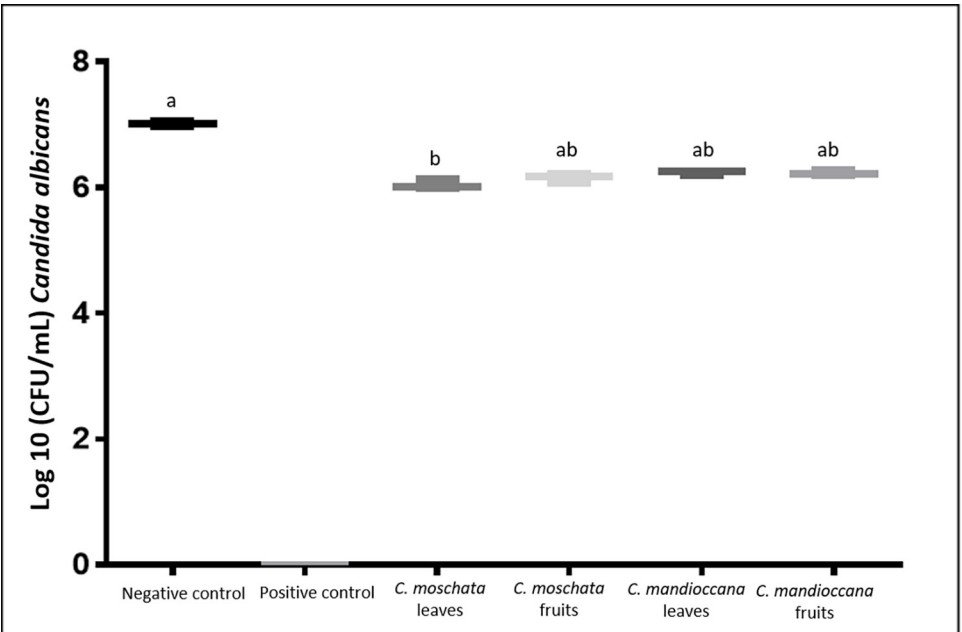

**Fig 6. CFU/mL values of *C. albicans* cells submitted to treatment with *Cryptocarya* extracts at a concentration of 0.030 g/mL (10 times the MIC value).**

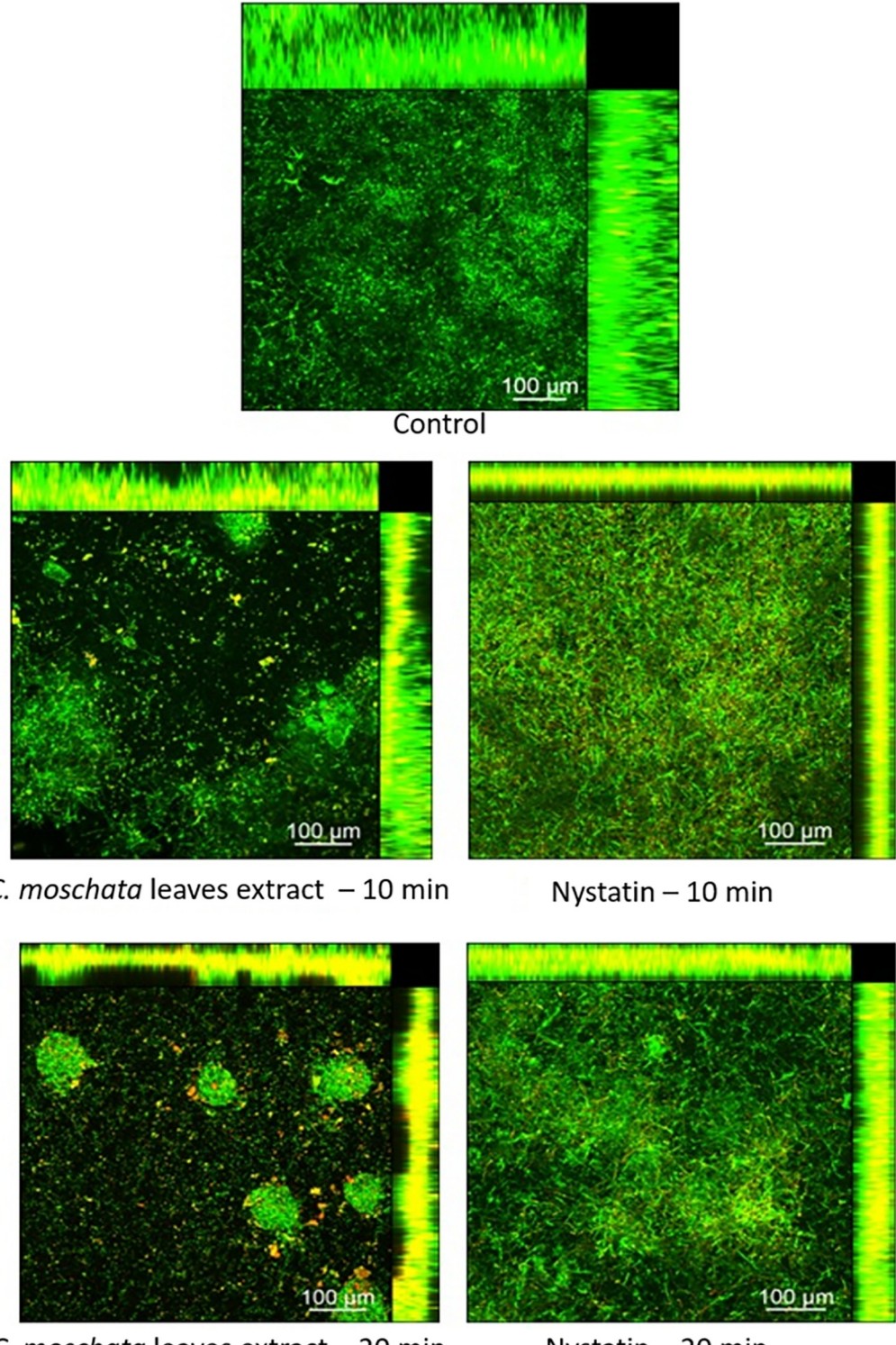

**Fig 7. CSLM of *C. albicans* adhered to acrylic resin samples after exposure to the *Cryptocarya* extracts (10 and 20 min) and nystatin (10 and 20 min) compared with the negative control group.** Samples Image capture was set for simultaneous observation of both green and red fluorescence.

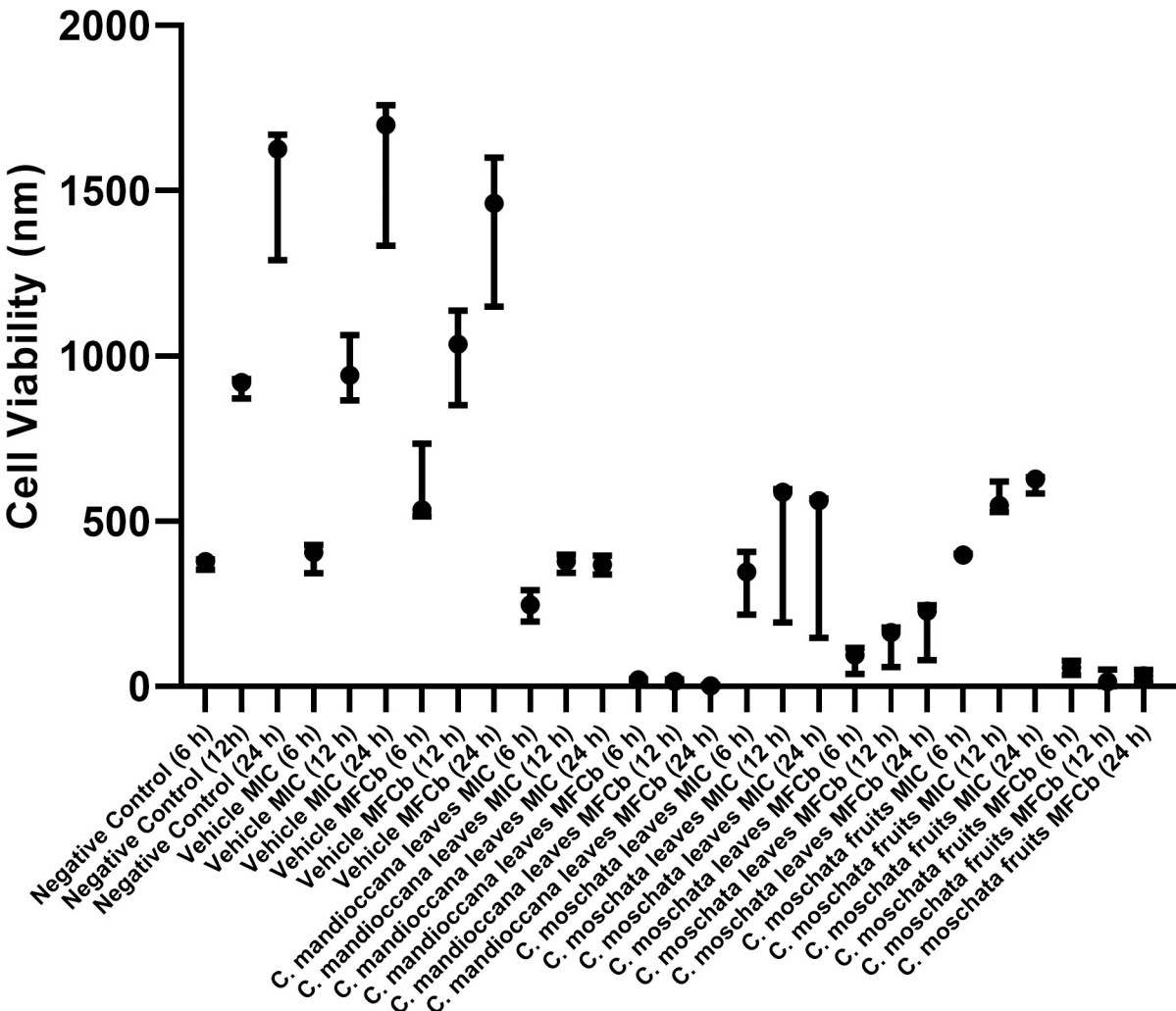

**Fig 8. Cell viability in fluorescence values.**

## Discussion

The null hypothesis of this study was rejected because the results showed that the extracts obtained from the leaves and fruits of *C. moschata* e *C. mandiocana* were able to reduce the number of colonies forming units and the cellular metabolism of the *C. albicans* biofilm formed on the acrylic resin samples, proving its antimicrobial effect. In addition, *Cryptocarya* species were cytotoxic to NOK cells.

The MFCb found was equal to 15 times to the MIC value (0,045 g/mL). This concentration completely inhibited the biofilm formed on the samples. It was also tested the concentration of 10 times to the MIC value (0,03 g/mL) for the extracts from leaves and fruits of *C. moschata* e *C. mandiocana*, however this concentration did not show a statistically significant reduction on *C. albicans* biofilm. These results are contrary to the ones reported by Maciel et al. [22], who used essential oils obtained from *C. aschersoniana* tested against filamentous fungi and yeast isolates. Nevertheless, the specie did not have antifungal activity against *C. albicans* isolates. However, Maciel et al. [22] evaluated essential oils, whereas, in the present study, the hydroalcoholic extracts of the species were evaluated.

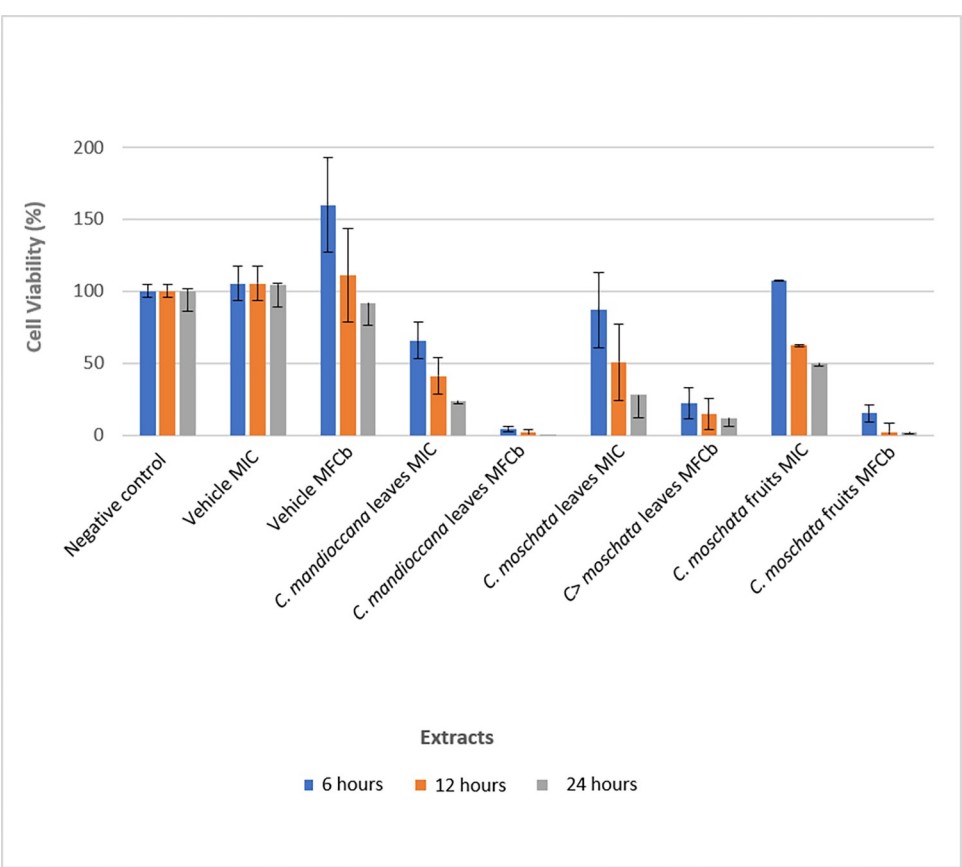

**Fig 9. Percentage of viable cells compared to the control group.**

Another aspect important to be discussed is the composition of the extracts once that this can be associated with their mechanism of action. In the UHPLC-DAD-MS analysis, the aporphine alkaloids, menisperine and xanthoplanine were detected in leaves and fruits of *C. moschata* and *C. mandioccana*, but not in leaves of *C. saligna*. Additionally, some styrylpyrones were detected. Based on previously reported phytochemical and pharmacological studies, the genus *Cryptocarya* are known to be prolific producers of flavonoids, chalcones, lactones, α-pyrones and mainly alkaloids with varied biological activities [23–32]. Zonaro [33] carried out a study with the purpose of identifying compounds present in three Brazilian species of *Cryptocarya* (*C. mandioccana*, *C. moschata* and *C. botelhensis*). Chromatographic method (HPLC-DAD-MS) was performed to analyze the hydroalcoholic extract of the leaves of *Cryptocarya* species, using ethanol as an organic phase. It was possible to identify that the classes of metabolites present in the samples were alkaloids, flavonoids and styrylpyrones. Another important substance is cryptomoscatone, which was isolated by Cavalheiro and Yoshida [23] from the bark of the Brazilian tree *C. moschata*, together with other representative structures. The compounds found by several studies can be related to the biological activities of the extracts, such as anticancer, antimicrobial, antifungal, insecticide, among others. Recently, Huang et al. [31, 32] reported the antimicrobial effect of flavonoids extracted from *C. concinna*. It is known that the flavonoids have different pharmacological properties, including antimicrobial and antioxidant activities. Likewise, Nasrullah et al. [34] also demonstrated the antimicrobial effect of alkaloids extracted from *C. nigra*. Regarding styrilpyrones, a study by Reddy et al. [35] demonstrated that these compounds are biologically active substances that have

anticancer, antioxidant, antifungal and antiviral activities. Additionally, menisperine exhibited moderate activity against *C. albicans* compared to the positive control [36]. The components derived from the *Cryptocarya* species can explain the antifungal effect found in the present study. However, the cell killing mechanism remains unclear.

It is important to highlight that the results of the present study are promisors, since that the extracts had antimicrobial effects in biofilms, which are much more resistant than planktonic cells. Biofilms, depicted as structured communities of microorganisms adhered to a biotic or abiotic surface and surrounded by an extracellular polymeric matrix [37], are produced by different species of *Candida*, varying in relation to production, composition and metabolic activity [38, 39]. Among these species, *C. albicans* is considered the one with the greatest capacity for biofilm production. The metabolic activity of a biofilm is important in terms of pathogenicity, as it is indicative of cellular growth and reproduction of microorganisms, production of toxic substances and greater antimicrobial tolerance [39]. In addition, when tested in biofilm models developed on acrylic surfaces, exhibited resistance to the antifungal agents nystatin and fluconazole [40], frequently used in the treatment of candidiasis. All the aforementioned reasons reinforce the relevance of our results.

The CLSM images are in accordance with the test results described above. Although the quantitative analysis showed predominance of live cells, there was a great difference between the control group (R = 10.8) and the other groups: extract 10 minutes (R = 4.6), extract 20 min (R = 2.2), nystatin 10 min (R = 2.1) and nystatin 20 min (R = 3,4), showing a decrease in the number of cells in the experimental groups and in the positive control.

Regarding the results of the cytotoxicity on NOK, all extracts decreased cell viability compared to the control group. When the results were converted into percentages and compared with the control group, it was found that the extracts in biofilm concentration were considered intensely cytotoxic, at all times. In MICs, extracts ranged from non-cytotoxic to intensely cytotoxic. These results are in accordance with other studies. Feng et al. [41] reported a moderate cytotoxicity of some constituents of *C. maclurei* on cancer cells. Huang et al. [31, 32], who tested the antimicrobial activity and cytotoxicity of flavonoids from *C. concinna*, observed that some compounds exhibited moderate cytotoxic activity. Recently, Xiong et al. [42] observed that some of the constituents from *C. impressinervia* exhibited significant cytotoxic activities in five human cancer cell lines. Cryptocarione, one of the components derived from the *Cryptocarya* species, has been reported as an agent responsible for its antiproliferative effect [29], which can explain the cytotoxicity of the results found here. The cell killing mechanism includes the ROS generation, mitochondrial depolarization, and DNA damage [29]. In addition, Andrade et al. [43] reported that the essential oil from *C. aschersoniana* leaves showed high toxicity against mice peritoneal macrophages, because the oil, due to its apolar character, penetrates into cells and does not act on specific cell receptors, causing cell death by necrosis and apoptosis. Cytotoxicity can also be explained by the presence of aporphine alkaloids detected in the UHPLC-DAD-MS analysis. Makarasen et al. [44] verified the cytotoxic activity of five alkaloids and concluded that some of them were cytotoxicity. The cytotoxic effects presented here should be viewed with caution as the test was performed on a cell monolayer. Considering that, further studies should be carried out in vivo models, increasing the reliability of the results obtained, emphasizing the good results found against mature biofilms of *C. albicans*.

In conclusion, the antimicrobial activity of the extracts of *Cryptocarya* in the control of *C. albicans* biofilm present in samples of denture base acrylic resin was confirmed. However, all extracts evaluated showed toxicity on the NOK cells. For this, in vivo studies are need to confirm the adverse effects on the tissue. Despite the existence of some studies [16, 17] considering the pharmacological effects of *Cryptocarya* species, few studies evaluated the antimicrobial

effect against *Candida* spp. biofilm [45]. Thus, the results of this study are important mainly for providing a low-cost, accessible and non-synthetic option for the treatment of oral fungal infections such as DS.

## Author Contributions

**Conceptualization:** Alberto José Cavalheiro, Janaina Habib Jorge.

**Data curation:** Jacqueline de Oliveira Zoccolotti.

**Formal analysis:** Jacqueline de Oliveira Zoccolotti, Túlio Morandin Ferrisse.

**Investigation:** Jacqueline de Oliveira Zoccolotti, Camilla Olga Tasso.

**Methodology:** Jacqueline de Oliveira Zoccolotti, Camilla Olga Tasso, Túlio Morandin Ferrisse.

**Project administration:** Janaina Habib Jorge.

**Visualization:** Alberto José Cavalheiro.

**Writing – original draft:** Jacqueline de Oliveira Zoccolotti, Alberto José Cavalheiro, Janaina Habib Jorge.

**Writing – review & editing:** Alberto José Cavalheiro, Beatriz Ribeiro Ribas, Janaina Habib Jorge.

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
