## [Decision Letter · Decision Letter 0]

31 Aug 2021

PONE-D-21-21992

Antimicrobial efficacy and biocompatibility of extracts from Cryptocarya species

PLOS ONE

Dear Dr. Jorge,

Thank you for submitting your manuscript to PLOS ONE. After careful consideration, we feel that it has merit but does not fully meet PLOS ONE’s publication criteria as it currently stands. Therefore, we invite you to submit a revised version of the manuscript that addresses the points raised during the review process.

The reviewers agree that the manuscript presents new information that may be of interest. However, there is detail that is missing from materials and methods that must be addressed. Please look at the comments of both reviewers. Additionally statistics need to be included. 

We look forward to receiving your revised manuscript.

Kind regards,

Joy Sturtevant

Academic Editor

PLOS ONE

Journal Requirements:

“This study was financed in part by the Coordenação de Aperfeiçoamento de Pessoal de Nível Superior – Brasil (CAPES) – Finance Code 88887.513643/2020-00.”

Reviewers' comments:

Reviewer's Responses to Questions

**Comments to the Author**

1. Is the manuscript technically sound, and do the data support the conclusions?

Reviewer #1: Yes

Reviewer #2: Yes

2. Has the statistical analysis been performed appropriately and rigorously? 

Reviewer #1: Yes

Reviewer #2: Yes

3. Have the authors made all data underlying the findings in their manuscript fully available?

Reviewer #1: Yes

Reviewer #2: Yes

4. Is the manuscript presented in an intelligible fashion and written in standard English?

Reviewer #1: Yes

Reviewer #2: Yes

5. Review Comments to the Author

Reviewer #1: The paper describes the fungicidal effect of Cryptocarya spp extracts on planktonic form and biofilm grown onto dental resin of Candida albicans. The paper also describes the cytocompatibility of the extract on oral keratinocytes.

The Authors tested C. moschata, C. saligna and C. mandioccana solutions at various concentrations in comparison with nystatin solution at 100,000 IU /mL and phosphate-buffered saline (PBS) solution.

The results showed the ability of the extract to inhibit the fungal growth but at the same time they revealed its cytotoxicity. C. saligna had no effect on C. albicans.

The paper is clear, and the experiments correctly done.

However, the paper is not fully informative, and the composition of the extract should be determined to understand the differences among the different Cryptocarya species and sources (leaves, nuts etc)

Figures should be included indication of statistical significance where relevant.

Reviewer #2: spp is not written in italics and as it is an abbreviation, add a period

The presentation of the extract preparation is poorly described. Was the plant material dried? If so, how, where, for how long and at what temperature? What is the concentration of the hydroalcoholic mixture for the extraction? To remove the solvent by lyophilization, first the ethanol must be removed with the aid of a rotaevaporator and then the residual water can be frozen and lyophilized. Was it possible to freeze the hydroalcoholic extract obtained to be later lyophilized?

In the statement "The crude extracts were weighed and aliquots containing adequate but were then diluted in water with ethanol in the proportions of interest", can a hydroalcoholic mixture be used to verify the antimicrobial action? Wouldn't the presence of alcohol be toxic? Therefore, action is to be expected in rehearsals and hence it is not valid.

For the preparation of Cryptocarya species concentrations, what does the Cryptocarya extracts 100% mean? Was undiluted extract used?

6. PLOS authors have the option to publish the peer review history of their article (what does this mean?). If published, this will include your full peer review and any attached files.

Reviewer #1: No

Reviewer #2: No

---

## [Author Response · Author response to Decision Letter 0]

17 Nov 2021

Journal Requirements:

Please remove any funding-related text from the manuscript and let us know how you would

like to update your Funding Statement. 

Response: The funding-related text from manuscript was removed and we confirm that “The funders had no role in study design, data collection and analysis, decision to publish, or preparation of the manuscript.”

Comments to the Author

Reviewer #1: The paper describes the fungicidal effect of Cryptocarya spp extracts on planktonic form and biofilm grown onto dental resin of Candida albicans. The paper also describes the cytocompatibility of the extract on oral keratinocytes. The Authors tested C. moschata, C. saligna and C. mandioccana solutions at various concentrations in comparison with nystatin solution at 100,000 IU /mL and phosphate buffered saline (PBS) solution. The results showed the ability of the extract to inhibit the fungal growth but at the same time they revealed its cytotoxicity. C. saligna had no effect on C. albicans. The paper is clear, and the experiments correctly done. 

Firstly, we would like to thank you for your considerations on the manuscript. All questions and considerations have been answered, as follows:

- The paper is not fully informative, and the composition of the extract should be determined to understand the differences among the different Cryptocarya species and sources (leaves, nuts etc).

Response: The analysis of plant extracts by UPLC-DAD-MS was performed and details about the methodology were added as suggested.

- Figures should be included indication of statistical significance where relevant.

Response: The indication of statistical significance as included as suggested.

Reviewer #2: 

Firstly, we would like to thank you for your considerations on the manuscript. All questions and considerations have been answered, as follows:

- spp is not written in italics and as it is an abbreviation. 

Response: The abbreviation was corrected in all the text as suggested.

- The presentation of the extract preparation is poorly described. Was the plant material dried? If so, how, where, for how long and at what temperature? What is the concentration of the hydroalcoholic mixture for the extraction? To remove the solvent by lyophilization, first the ethanol must be removed with the aid of a rot evaporator and then the residual water can be frozen and lyophilized. Was it possible to freeze the hydroalcoholic extract obtained to be later lyophilized?

Response: Details about the methodology were added as suggested.

- In the statement "The crude extracts were weighed and aliquots containing adequate but were then diluted in water with ethanol in the proportions of interest", can a hydroalcoholic mixture be used to verify the antimicrobial action? Wouldn't the presence of alcohol be toxic? Therefore, action is to be expected in rehearsals and hence it is not valid.

Response: For microbiological analyses, the crude extracts were weighed and aliquots and, then, diluted in phosphate bufferid saline (PBS) with ethanol (5%) in the proportions of interest. This concentration is not toxic to oral cells*.

*Calderón-Montaño JM, Jiménez-Alonso JJ, Guillén-Mancina E, Burgos-Morón E, López-Lázaro M. A 30-s exposure to ethanol 20% is cytotoxic to human keratinocytes: possible mechanistic link between alcohol-containing mouthwashes and oral cancer. Clin Oral Investig. 2018 Nov;22(8):2943-2946.

- For the preparation of Cryptocarya species concentrations, what does the Cryptocarya extracts 100% mean? Was undiluted extract used?

Response: The extract 100% is the undiluted extract.

Best regards.

Sincerely,

Janaina Habib Jorge

---

## [Decision Letter · Decision Letter 1]

14 Dec 2021

Antimicrobial efficacy and biocompatibility of extracts from Cryptocarya species

PONE-D-21-21992R1

Dear Dr. Jorge,

We’re pleased to inform you that your manuscript has been judged scientifically suitable for publication and will be formally accepted for publication once it meets all outstanding technical requirements.

Kind regards,

Joy Sturtevant

Academic Editor

PLOS ONE

Additional Editor Comments (optional):

Reviewers' comments:

Reviewer's Responses to Questions

**Comments to the Author**

1. If the authors have adequately addressed your comments raised in a previous round of review and you feel that this manuscript is now acceptable for publication, you may indicate that here to bypass the “Comments to the Author” section, enter your conflict of interest statement in the “Confidential to Editor” section, and submit your "Accept" recommendation.

Reviewer #1: All comments have been addressed

Reviewer #2: All comments have been addressed

2. Is the manuscript technically sound, and do the data support the conclusions?

Reviewer #1: Yes

Reviewer #2: Yes

3. Has the statistical analysis been performed appropriately and rigorously? 

Reviewer #1: Yes

Reviewer #2: Yes

4. Have the authors made all data underlying the findings in their manuscript fully available?

Reviewer #1: Yes

Reviewer #2: Yes

5. Is the manuscript presented in an intelligible fashion and written in standard English?

Reviewer #1: Yes

Reviewer #2: Yes

6. Review Comments to the Author

Reviewer #1: Thank you for your replies. Paper is improved after revision. Extarcts from natural sources are interesting new therapeutic tools in many application including, of course, antimicrobial treatment

Reviewer #2: The suggestions given for the article "Antimicrobial efficacy and biocompatibility of extracts from Cryptocarya species" were met. So the manuscript is ok to be published.

7. PLOS authors have the option to publish the peer review history of their article (what does this mean?). If published, this will include your full peer review and any attached files.

Reviewer #1: No

Reviewer #2: No

---

## [Editor Report · Acceptance letter]

21 Dec 2021

PONE-D-21-21992R1 

Antimicrobial efficacy and biocompatibility of extracts from *Cryptocarya* species 

Dear Dr. Jorge:

I'm pleased to inform you that your manuscript has been deemed suitable for publication in PLOS ONE. Congratulations! Your manuscript is now with our production department. 

Kind regards, 

on behalf of

Dr. Joy Sturtevant 

Academic Editor

PLOS ONE